# Evaluation of Long-Term Adaptive Immune Responses Specific to SARS-CoV-2: Effect of Various Vaccination and Omicron Exposure

**DOI:** 10.3390/vaccines12030301

**Published:** 2024-03-13

**Authors:** Hyunhye Kang, Jin Jung, Geon Young Ko, Jihyun Lee, Eun-Jee Oh

**Affiliations:** 1Department of Laboratory Medicine, Seoul St. Mary’s Hospital, College of Medicine, The Catholic University of Korea, Seoul 06591, Republic of Korea; hyunhye@cmcnu.or.kr (H.K.); jiinj@catholic.ac.kr (J.J.); 2Research and Development Institute for In Vitro Diagnostic Medical Devices, College of Medicine, The Catholic University of Korea, Seoul 06591, Republic of Korea; 3Department of Biomedicine and Health Sciences, Graduate School, The Catholic University of Korea, Seoul 06591, Republic of Korea; geonyoung0107@catholic.ac.kr (G.Y.K.); onion1002@catholic.ac.kr (J.L.)

**Keywords:** SARS-CoV-2, vaccine, humoral immunity, cellular immunity, breakthrough infection, long-term, Omicron

## Abstract

The immune response to severe acute respiratory syndrome coronavirus 2 (SARS-CoV-2) becomes increasingly complex as individuals receive different combinations of vaccine doses and encounter breakthrough infections. Our study focused on the immunogenicity observed over a two-year period in healthy individuals who completed a two-dose series and then experienced booster and/or Omicron infection. In June 2023, we recruited 78 healthcare workers who had previously participated in clinical research initiated in March 2021 at a single medical center in South Korea. At 1, 5, 11, and 25 months after a second dose, we assessed SARS-CoV-2–specific humoral and cellular immune responses. Longitudinal monitoring revealed a significant decline in humoral immunity levels after the second vaccine dose, followed by a substantial increase post-third vaccination or breakthrough infection. In contrast, stable cellular immune responses were consistently observed, with peak humoral and cellular immune measures reached at 25 months after the second dose. Among infection-naïve participants, three-dose vaccinated individuals had decreased neutralizing activity against wild-type (WT) and negative activities against Omicron subvariants BA.2 and BA.4/5, whereas those who received a fourth dose of bivalent BNT had significantly increased neutralizing activity (*p* < 0.05). All immune metrics tended to increase as the number of vaccine doses increased. Among participants with 4-exposure, homologous vaccination (mRNA × 4) led to higher humoral immunity, whereas heterologous vaccination (ChAd × 2/mRNA × 2) induced stronger cellular responses against multiple SARS-CoV-2 variants by enzyme-linked immunospot assays (*p* < 0.05). Immune responses from bivalent vaccines or Omicron infection did not show statistically significant differences among exposure number-matched participants (*p* > 0.05). Omicron exposure significantly increased cross-neutralizing activity, but magnitude of cellular immunity was not significantly altered by Omicron exposure. Our longitudinal study highlights the evolving complexity of SARS-CoV-2 immune responses, showing enhanced immunity with multiple vaccine doses and robust cellular responses from heterologous vaccination. These findings emphasize the need for ongoing surveillance to optimize vaccination strategies against emerging variants.

## 1. Introduction

Four years have passed since the World Health Organization (WHO) declared the severe acute respiratory coronavirus-2 (SARS-CoV-2) outbreak a Public Health Emergency of International Concern on 31 January 2020 and a global pandemic on 11 March 2020. Over the course of the pandemic, various vaccine platforms have been introduced, playing a crucial role in preventing infection [1,2]. However, vaccine-induced immune responses have waned rapidly, and the emergence of immune-evasive variants has contributed to the increasing number of breakthrough infections [3,4,5]. In particular, vaccines provided limited protection against the currently circulating Omicron variant. In response to this variant, a third booster dose was administered, leading to a significant increase in neutralizing capacity. However, the neutralizing antibody titers also showed a rapid decline within three months after the third dose, particularly against sublines BA.2.12.1 and BA.4/5 [6]. Consequently, each individual’s current immune profile is shaped by both vaccine-induced and infection-induced immune responses.

Previous studies have suggested that repeated exposure to the viral antigen, whether through vaccination, infection, or both, leads to the development of high-avidity cross-neutralizing activity [7,8,9]. The synergistic impact of vaccination and infection induces a particularly potent effect known as hybrid immunity. This hybrid immunity demonstrates enhanced effectiveness compared to immunity solely induced by vaccination, resulting in stronger neutralization and better adaptive immunity [5,10]. Regarding the cellular immune response, it appears to be more enduring and consistent, as opposed to the rapid rise and fall observed in the antibody response [11].

However, there remains a gap in our understanding of current immune profiles following diverse vaccination and breakthrough infection histories. In addition, there is heterogeneity amongst vaccinated individuals influenced by factors such as the total number of antigen exposures from either infection or vaccination, vaccine strategies and the specific variant responsible for the infection.

Previous longitudinal studies of SARS-CoV-2–specific immune responses have covered a period of up to one year after the completion of the primary two-dose series [5,11,12,13,14,15,16] or up to 20 months after infection [17]. However, the specific characteristics of long-term immune response metrics in infection-naive individuals remain largely unexplored. Therefore, the need for additional vaccination and the prioritization of recipients requires further study and consideration.

This study is part of a longitudinal investigation involving naïve participants who initially received two doses of vaccines and subsequently either received booster vaccinations, recovered from Omicron infections, or experienced both. We performed a comprehensive analysis of the humoral and cellular immune response to both wild-type (WT) and the Omicron subvariants (BA.1., BA.2., and BA.4/5) two years after the primary series vaccination. This study aims to analyze the immune profiles associated with SARS-CoV-2–specific adaptive immune responses in vaccinated individuals. We investigated the impact of diverse antigen exposure histories, including variations in vaccine doses, types, and Omicron exposure through breakthrough infections or bivalent vaccinations.

## 2. Materials and Methods

### 2.1. Enrolment and Sample Collection

In March 2021, we enrolled a total of 359 infection-naive healthcare workers at Seoul St. Mary’s Hospital and initiated the longitudinal monitoring of SARS-CoV-2–specific immune responses after a primary series of two-dose vaccination. All participants began their initial SARS-CoV-2 vaccination in March 2021 and completed the primary two-dose series. They received either mRNA vaccine BNT162b2 (Pfizer-BioNTech, hereafter referred to as BNT) or mRNA-1273 (Moderna) or were given the vector-based ChAdOx1 nCoV-19 (AstraZeneca, hereafter referred to as ChAd). Subsequently, in June 2023, 78 individuals from the original cohort consented to participate in this follow-up study.

In this study, we assessed the long-term SARS-CoV-2–specific immune responses following booster vaccinations and/or Omicron breakthrough infection at 25 months (T4) after completing the primary two-dose vaccination series. This study was approved by the Institutional Review Board of Seoul St. Mary’s Hospital (KC23TISI0183). All participants provided written informed consent. Additionally, they granted permission for the use of their immunogenicity test results obtained at 1 month (T1), 5 months (T2) and 11 months (T3) after completing the primary two-dose vaccination series from previous studies (KC21DIST0174). Participants were asked to complete a medical history questionnaire that included information on demographics, underlying medical conditions, SARS-CoV-2 vaccination history, and any prior experiences with breakthrough infections. Breakthrough infection was confimred based on the available reverse transcription polymerase chain reaction (RT-PCR) on the electronic medical record and anti-nucleocapsid (N) antibody positivity. Infection-naive was defined based on negative survey and negative anti-N test results.

### 2.2. SARS-CoV-2–Specific Humoral Immune Response Assays

Serum samples from participants were analyzed using the Elecsys Anti-SARS-CoV-2 assay (Roche Diagnostics, Basel, Switzerland) to detect binding antibodies to the receptor-binding domain (RBD) of the spike protein (anti-S/RBD) and anti-N, as per the manufacturer’s instructions. Quantification of the anti-S/RBD antibody was performed in units per mL (U/mL), and binding antibody units per mL (BAU/mL) were calculated based on the WHO International Standard for anti-SARS-CoV-2 immunoglobulin. A cut-off value of 0.8 U/mL for anti-S/RBD was utilized, as recommended by the manufacturer. The result for anti-N is presented as a cut-off index (COI), where COI > 1.0 indicates a reactive result. The reactive anti-N result was used to detect asymptomatic breakthrough infections that might have gone unreported by participants.

The SARS-CoV-2 surrogate virus neutralization test (sVNT) (GenScript cPassTM, Piscataway, NJ, USA) was conducted following previously described procedures to evaluate the binding inhibition value of neutralizing antibodies [18]. Serum samples from participants, along with positive and negative controls, were diluted 1:10 with dilution buffer. They were then mixed with horseradish peroxidase–conjugated recombinant SARS-CoV-2 RBD solution provided in the abovementioned GenScript test and incubated at 37 °C for 30 min. Following this, the mixtures were incubated for 15 min at 37 °C in a capture plate precoated with human angiotensin-converting enzyme 2 (hACE2) protein in the GenScript kit. After washing, tetramethylbenzidine (TMB) solution in the sVNT assay kit was added, and the plate was incubated in darkness at room temperature for 15 min. The reaction was halted with stop solution, and absorbance was measured at 450 nm using an enzyme-linked immunosorbent assay (ELISA) microplate reader. Following the manufacturer’s instructions, a cut-off of ≥30% inhibition was considered indicative of positive neutralizing activity. Each sample was tested for SARS-CoV-2 WT, Omicron BA.2 (BA.2), and Omicron BA.4 and BA.5 (BA.4/5) spike proteins.

### 2.3. SARS-CoV-2–Specific Cellular Immune Response Assays

The cellular immune response was assessed utilizing the enzyme-linked immunospot (ELISpot) assay. Peripheral blood mononuclear cells (PBMCs) were stimulated with PepTivator SARS-CoV-2 S1 peptide pools of WT and Omicron subvariants BA.1, BA.2, and BA.5 (Miltenyi Biotec, Bergisch Gladbach, Germany). These peptide pools consisted of 15-mer sequences with an 11 amino acid (aa) overlap, spanning aa 1-692. The S1 domain includes the RBD and N-terminal domain, well-known targets of neutralizing antibodies [9]. The 96-well plates were coated overnight at 4 °C with anti-interferon (IFN)-γ monoclonal capture antibodies sourced from Human IFN-γ ELISpot kits (BD Biosciences, San Jose, CA, USA). After adding blocking solution, 2.5 × 10^5^ cells/mL per well were stimulated with antigens. Plates were maintained overnight at 36 °C in a CO_2_ incubator. After multiple washes, AEC substrate was added for 25 min and kept in dark lighting at room temperature. The IFN-γ spot forming cells were enumerated using the AID ELISpot reader system (Autoimmun Diagnostika GmbH, Strasburg, Germany). Results were quantified as IFN-γ spot forming units (SFU)/2.5 × 10^5^ PBMCs.

### 2.4. Statistical Analysis

Continuous data are presented as median with interquartile ranges (IQR). Paired samples were compared by the Wilcoxon sign-rank test. Statistical comparisons between immune measures among different groups were conducted using the Mann–Whitney U test and the Kruskal–Wallis test, with Dunn’s multiple comparisons test employed post hoc as necessary, depending on the number of comparisons required. Categorical data were presented as counts and percentages and were analyzed using either the chi-square or Fisher’s exact test. The association between test results was evaluated using the Spearman rank correlation test. Data analysis and visualization were carried out using Prism version 10.0.2 for Windows (GraphPad, San Diego, CA, USA) and MedCalc statistical software version 20.114 (MedCalc Software Ltd., Ostend, Belgium). A significance threshold of *p* < 0.05 (two-tailed) was applied for all statistical analyses.

## 3. Results

### 3.1. Participant Characteristics

The characteristics of all participants were described according to the number of antigen exposures (Figure 1 and Appendix A). Each vaccination or breakthrough infection event was considered as one exposure. The 78 enrolled participants were categorized into three groups based on the number of antigen exposures as follows: 3-exposure (*n* = 8), 4-exposure (*n* = 50) and 5 or more exposures (5+exposure) (*n* = 20). The 5+exposure group included 17 participants with 5-exposure and 3 participants with 6-exposure. Among the participants, 69 (88.5%) participants experienced breakthrough infections, and nine (11.5%) participants remained infection-naïve. Breakthrough infections were reported between January 2022 and January 2023, coinciding with the Omicron pandemic wave in South Korea [19,20,21,22]. Since all reported breakthrough infections occurred during the Omicron wave, these events were considered to be Omicron exposures. Three participants in the 4-exposure group and one participant in the 5+exposure group did not experience a breakthrough infection, instead receiving a bivalent vaccine targeting the Omicron BA.4/5 subvariant. Participants who received a bivalent vaccine targeting the Omicron BA.4/5 subvariant were considered to have been exposed to Omicron. Overall, all except for five participants in the 3-exposure group were exposed to Omicron either by infection or bivalent vaccination.

The median age of participants was 46 years old (37–52 [IQR]), and 73.1% (57/78) of them were female. Of those, 25.6% (20/78) reported having medical comorbidities, with the most common ones being hypertension (9/78) followed by diabetes mellitus (6/78). All participants had completed the two dose primary series of vaccination. Among them, 82.1% (64/78) had received ChAd, while 18.0% (14/78) had received mRNA vaccines (12 BNT, and 2 mRNA-1273). A total of 94.9% (74/78) participants received at least one booster dose, with 73.1% (57/78) receiving a total of three doses, 20.5% (16/78) receiving four doses, and 1.3% (1/78) receiving five vaccine doses.

Of the 69 participants with breakthrough infections, eight individuals (8/69, 11.6%) have been infected twice, and one participant (1/69, 1.4%) reported three infections. Based on the primary vaccine series, 87.5% (56/64) of those primed with ChAd and 92.9% (13/14) of those with mRNA vaccines experienced breakthrough infections. When comparing nine naïve and 69 infected participants, no significant differences were found in baseline demographics, primary vaccine schedule, or the number of vaccine doses (Appendix A). The immune-conferring scenario of the participants is described in Figure 1. The final sample was collected approximately 25 months (T4) after second dose administration. The median intervals for specimens collected at 1 month (T1), at 5 months (T2) and at 11 months (T3) were 26.0 (23.0–29.0), 161.0 (160.0–165.0) and 336.0 (332.3–338.0) days after the second dose of vaccine, respectively (Appendix A).

### 3.2. Longitudinal Kinetics of Humoral and Cellular Immune Response

The levels of anti-S/RBD and neutralization activity against the ancestral spike significantly declined from T1 to T2 (*p* < 0.0001; Wilcoxon test) post the second dose vaccination (Figure 2A,B). Then, a significant increase was observed during the period from T2 to T3 (anti-S/RBD, 17.5-fold increase; sVNT, 2.0-fold increase; *p* < 0.0001 by Wilcoxon test) after receiving the third dose vaccination (37/59) or third dose vaccination followed by breakthrough infection (22/59). Between T3 and T4, the majority (93.6%) of participants experienced breakthrough infections and/or booster vaccination. Despite these immune-conferring events, the rise in anti-S/RBD titer did not reach statistical significance (2.1-fold increase, *p* = 0.0749; Wilcoxon test), and while the neutralizing activity revealed a statistically significant increase, the change was modest (median increase from 97.0% to 97.9%, *p* = 0.0015; Wilcoxon test). Longitudinal cellular immune responses were also analyzed. The SARS-CoV-2–specific cellular immune response was maintained by stable kinetics throughout the investigated period compared to the humoral immune response. A significant 1.4-fold increase in the magnitude of T-cell immunity to S1 peptide stimulation was observed between T2 and T3 (*p* = 0.0001; Wilcoxon test) (Figure 2C).

Interestingly, the highest immune measures were observed at T4, a median of 575 days after the last vaccination. Antibody titers directing S/RBD increased 7.9-fold (*p* < 0.0001; Wilcoxon test), neutralizing activity increased 1.1-fold (*p* < 0.0001; Wilcoxon test), and IFN-γ SFU increased 1.9-fold (*p* = 0.0026; Wilcoxon test) compared to T1 results, which represent the peak response after the second dose administration.

### 3.3. Effect of the Number of Vaccine Doses Stratified by Number of Breakthrough Infection on SARS-CoV-2–Specific Adaptive Immune Response

We evaluated the effect of the number of vaccine doses on the long-term immune response at T4, stratified by number of breakthrough infections. Across all groups categorized by the number of infections, there was a trend of increasing immunogenicity parameters in line with the number of vaccine doses administered (Figure 3A–H). The observed anti-S/RBD titer aligned with this overall trend, although statistical significance was found between participants who received three doses and those who received four doses among participants infected once (*p* < 0.05) (Figure 3A). Notably, five infection-naïve participants who received three doses of vaccination, and had no prior exposure to Omicron through infection or vaccination, exhibited decreased neutralizing activity against WT and had negative (<30%) neutralizing activities against the Omicron subvariants, BA.2 and BA.4/5. However, infection-naive participants who received a fourth dose of bivalent BNT exhibited increased neutralizing activity against WT, BA.2, and BA.5 compared to naïve individuals who received three doses (*p* < 0.05; Mann–Whitney U test) (Figure 3B–D). Unlike the humoral response, there were no statistically significant differences observed in cellular responses based on the number of vaccine doses within the same infection frequency group (Figure 3E–H).

### 3.4. Effect of Heterologous versus Homologous Vaccination on SARS-CoV-2–Specific Adaptive Immune Response

Next, we compared the immunogenicity of heterologous and homologous vaccination schedules in participants who received three or more vaccinations. In our cohort, heterologous vaccinatons were defined as individuals primed with ChAd and boosted with BNT, while homologous vaccinations included participants vaccinated with mRNA, either BNT or mRNA-1273. When we stratified participants based on the total number of antigen exposures, the comparison of heterologous versus homologous vaccination was only available for those with 4-exposures; (ChAd × 2/mRNA × 2) (*n* = 38) versus. (mRNA × 4) (*n* = 11). Interestingly, participants vaccinated with homologous regimens exhibited significantly higher levels of anti-S/RBD (*p* = 0.0044; Mann–Whitney U test) and neutralizing activity against WT (*p* = 0.0001; Mann–Whitney U test), as well as BA.2 (*p* = 0.0011; Mann–Whitney U test) and BA.5 (*p* = 0.0018; Mann–Whitney U test) (Figure 4A,B). Conversely, those vaccinated with heterologous regimens displayed a greater magnitude of IFN-γ–producing cellular response against WT (*p* = 0.0582; Mann–Whitney U test), BA.1 (*p* = 0.0149; Mann–Whitney U test), BA.2 (*p* = 0.0199; Mann–Whitney U test), and BA.5 (*p* = 0.0760; Mann–Whitney U test) S1 peptide pools (Figure 4C).

### 3.5. Effect of Breakthrough Infection and Bivalent Vaccination Stratified by Number of Antigen-Exposure on SARS-CoV-2–Specific Adaptive Immune Response

Because both infection and vaccination induce virus-specific adaptive immune responses, we evaluated the effect of breakthrough infection and bivalent vaccination in individuals stratified by the number of total antigen exposures. In the 3-exposure group, individuals with breakthrough infection (2-vaccination + 1-breakthrough infection, 2Vac/1B) had an 11.8-fold increase in anti-S/RBD (*p* = 0.0714; Mann–Whitney U test) and neutralizing activity against WT (1.2-fold, *p* = 0.0357; Mann–Whitney U test), BA.2 (6.1-fold, *p* = 0.0357; Mann–Whitney U test), and BA.5 (6.8-fold, *p* = 0.0357; Mann–Whitney U test) compared to those without breakthrough infection (3-vaccination, 3Vac) (Figure 5A–D). Conversely, the magnitude of the cellular response was more pronounced in infection-naive participants, though none reached statistical significance (Figure 5E–H). In the 4- and 5+exposure groups, bivalent vaccines or breakthrough infection displayed immune responses that were not statistically different in participants matched for the number of exposures (4Vac vs. 3Vac/1B and 4+Vac/1B vs. 3+Vac/2+B) (Figure 5).

### 3.6. Cross-Immunity against WT and Omicron Variants in Groups Stratified by Number of Exposures

As the evolving SARS-CoV-2 strains are derived from the Omicron strains [23], we assessed the humoral and cellular immune reaction against the Omicron sublineages in groups stratified by the number of exposures. In all exposure groups, the highest neutralization activity was found against the WT spike. Within the 3-exposure group, five participants who never exposed to Omicron showed notable decreases in neutralizing activity against Omicron spikes BA.2 and BA.4/5 compared to WT. Conversely, groups with four or more exposures who had Omicron infection showed a largely retained neutralizing capacity against Omicron spikes BA.2 and BA.4/5, although not as potent as the WT (Figure 6A). In contrast to humoral immunity, ELISpot responses to various spike peptide pools (WT, BA.1, BA.2, and BA.5) showed no significant differences among individuals stratified by the number of exposures, representing cellular cross-immunity (Figure 6B).

To assess the impact of Omicron infection on cross-immunity, we examined the correlation of immune responses between WT and Omicron subvariants separately for infection-naive individuals (Appendix A) and individuals with breakthrough infections (Appendix A). The overall humoral immune response showed a strong correlation in both infection-naive (Spearman rho = 0.85–0.98) and breakthrough infected individuals (Spearman rho = 0.51–0.89). The cellular immune parameters showed a heterogeneous correlation in the infection-naive participants (Spearman rho = 0.15–0.87), whereas breakthrough-infected participants showed a better coordinated correlation of cellular response across different variants of peptide stimulation (Spearman rho = 0.61–0.84). Additionally, an inverse correlation was found between the humoral immune response compartment and cellular parameters in participants with breakthrough infections.

## 4. Discussion

Individual immune profiles are shaped by virus-specific vaccination and breakthrough infection. Understanding the current immune response is crucial for making future plans for controlling COVID-19. Given the dynamic nature of adaptive immune responses, the updated immune profiles reflecting various immunity-conferring events are required. This study offers a longitudinal monitoring spanning over two years after the completion of the primary two-dose vaccination. We evaluated present immune responses across groups with diverse immunogenic histories and identified individuals with diminished immune responses that may require prioritization in future vaccine plans.

Previous research has suggested that repeated antigenic stimuli contributes significantly to robust and broader humoral responses [7,8,10,13,15] in comparison to the declining humoral immune response observed in individuals who completed only two doses of BNT [14]. We also observed a robust humoral immune response following the third dose, consistent with the findings at T3 and T4. This contrasts with the rapid waning of binding antibody titers and neutralizing inhibition observed from T1 to T2 between median 26 to 161 days after the second dose administration. This finding supports previous observations that the third dose elicited the most potent humoral immune response [8,9,24,25]. Notably, an enhanced humoral response was evident in those exposed to Omicron. The stronger inhibition observed against the WT spike of SARS-CoV-2 compared to BA.2 and BA.4/5 among individuals exposed to Omicron can be attributed to the phenomenon of back-boosting associated with immune imprinting [26]. Contradictory findings exist regarding the impact of immune imprinting on SARS-CoV-2 vaccine effectiveness [27,28]. A recent study, however, provided an encouraging result suggesting that the BNT162b2 bivalent BA.4/5 vaccine elicited a broader neutralization activity, extending to currently circulating more immune-evasive Omicron sublineages such as BA.2.75.2, BA.4.6, BQ.1.1, and XBB.1 [29]. Similarly, the bivalent Omicron BA.1-containing mRNA-1273.214 vaccine (Moderna) was reported to induce broad neutralizing capacity against the alpha, beta, gamma, delta, and Omicron variants [30].

When specifically focusing on neutralization, individuals exposed to the Omicron exhibited an expanded breadth of response compared to the five infection-naïve participants in this study, whose SARS-CoV-2–specific immune response has been only induced by vaccination based on ancestral spikes. This finding might seem contradictory to a previous study [8] that reported three doses targeting the ancestral spike resulted in broader neutralizing activity to cover SARS-CoV-2 variants. Their evaluation was based on samples collected 48 days after the third dose, representing an early response. As our samples were collected a median of 744 days after second dose and 575 days after third dose vaccination, our results raise the question of whether cross-neutralization from three or more ancestral spike-targeting doses is less enduring than Omicron spike exposure. It is noteworthy that infection-naïve individuals vaccinated with bivalent BNT exhibited neutralizing activity comparable to their counterparts in the exposure number matched group.

Compared to the humoral immune response, the cellular immune response remained stable in response to vaccination and breakthrough infection in line with previous research [5,13,16,29]. However, our finding also revealed substantial increases in the magnitude of T-cell response after third exposure. This observation may indicate that the third antigenic stimuli have a substantial impact on the cellular immune response as well, suggesting that its role extends beyond influencing the duration and scope of the humoral immune response. An interesting finding was that the T cell magnitude induced by Omicron peptide pools was higher in infection-naïve individuals non-exposed to the Omicron. Initially the vaccine-induced immunity exhibits varying degrees of correlation across various immune metrics. Even infection-naïve individuals who were never exposed to the Omicron spike exhibited a degree of cross-reactive cellular immune response. Taken together, the augmented cross-reactive cellular immunity could potentially contribute to protect them against breakthrough infections. This finding is consistent with previous reports suggesting that durable and extensive cross-reactive cellular immunity may contribute to protection against infection [9,31].

In this context, it is noteworthy that an enhanced cellular response was induced in individuals who followed a heterologous vaccine schedule, comprising vector-based vaccines primed and boosted by a mRNA-based vaccine. Notably, all nine participants who remained non-infected during the observation period in this study had followed a heterologous vaccination schedule. One intriguing hypothesis is that this group may have benefited from a dual advantage: a stronger cellular response conferred by the vector-based vaccines and more robust humoral responses induced by the mRNA vaccines. However, since the heterologous vaccines did not yield statistically significantly higher rates of protection in this study, further investigation is required to confirm this hypothesis.

After breakthrough infection, a dissociation emerges between humoral immunity parameters and cellular compartment. Each component demonstrated a moderate to strong correlation within the compartment, yet a negative correlation emerges between humoral and cellular immune measures. The better correlated immune response among breakthrough infectees was previously described [5]. The inverse correlation can be explained by the hypothesis that individuals with a higher magnitude of T-cell adaptive immune response were protected from breakthrough infection, while those with lower responses experience breakthrough infections. Then, the infection itself prompted a rapid humoral response against the virus. Given the current context of widespread vaccination and breakthrough infections, depending solely on humoral immunogenicity measures as indicators of protection might not provide a comprehensive perspective for assessing vaccine effectiveness.

Our study has several limitations. As an observational study, we were unable to capture all possible combinations of the number of exposed antigens and the specific exposed variant, while the limited number of participants may affect the statistical power and reduce generalizability. Another limitation of our study is that we did not identify the causative variants in individuals with breakthrough infections through sequence analyses. However, the timing of the infection provides an indication of the likely circulating variant.

## 5. Conclusions

Our study assessed the continued immune response over a two-year period following the completion of the primary two-dose vaccination schedule, with and without booster vaccinations and/or breakthrough infection. Our longitudinal study highlights the evolving complexity of SARS-CoV-2 immune responses, showing enhanced immunity following multiple vaccine doses and robust cellular responses from heterologous vaccinations. These findings emphasize the need for ongoing surveillance to optimize vaccination strategies against emerging variants and to identify individuals to be potentially prioritized in future booster vaccination plans.

## Figures and Tables

**Figure 1 vaccines-12-00301-f001:**
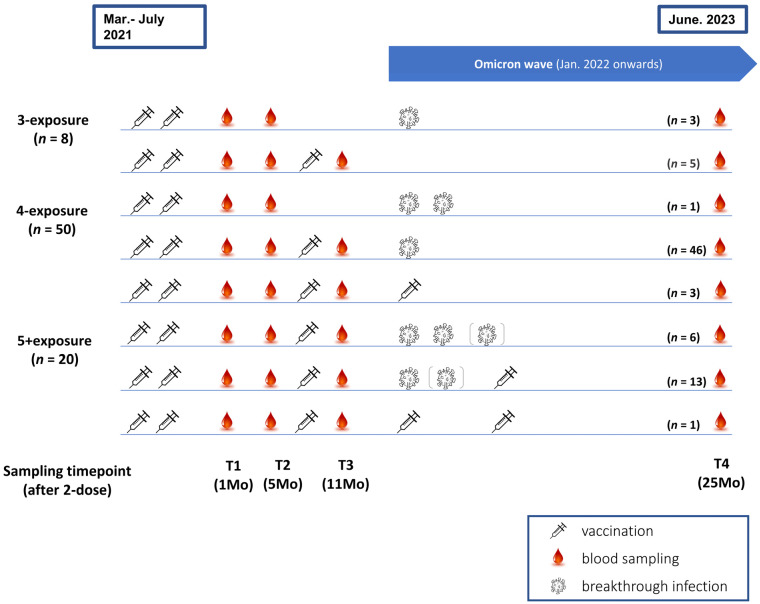
Timeline illustrating vaccination, infection and sampling timepoints of the study participants. The icons depicted in the figure represent the following: A syringe indicates one vaccine dose administration, blood droplets denote blood sampling, and a virus signifies breakthrough infection. Parentheses accompanying the virus figures indicate that some participants in the group experienced either a third or second infection.

**Figure 2 vaccines-12-00301-f002:**
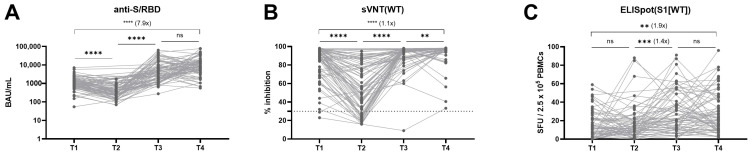
Longitudinal kinetics of SARS-CoV-2–specific anti-S/RBD (**A**), neutralizing activity to WT (**B**), and the magnitude of T cell response by IFN-γ ELISpot (**C**). All 78 participants of this study are presented as T4 results. The available previous measurements of immunogenicity parameters are presented as T1, T2, and T3, as defined in Appendix A. The individual dots represent an assay result at each sampling point. The horizontal dotted line in (**B**) indicates the manufacturer’s suggested cut-off, while ns, not significant, ** *p* < 0.01, *** *p* < 0.001, **** *p* < 0.0001 by Wilcoxon sign-rank test.

**Figure 3 vaccines-12-00301-f003:**
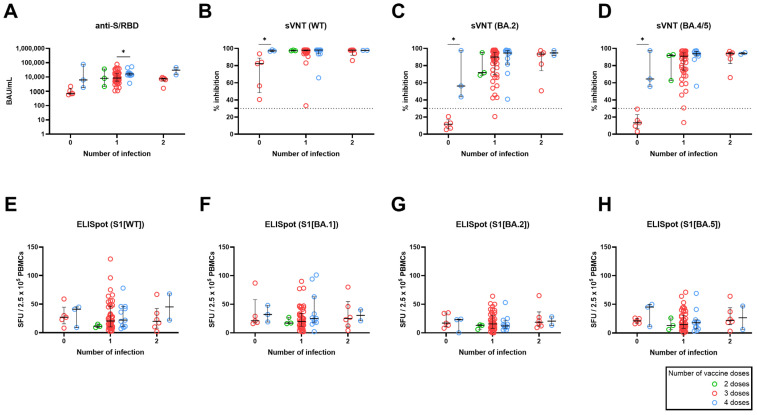
Effect of the number of vaccine doses on SARS-CoV-2–specific humoral and cellular immune response, stratified by the number of breakthrough infections. Humoral responses were measured using anti-S/RBD levels (**A**), neutralizing activity against WT (**B**), BA.2 (**C**), and BA.4/5 (**D**). The magnitudes of T-cell responses were measured using an enzyme-linked immunospot assay (**E**–**H**). The number of vaccine doses is color-coded, as indicated at the right corner. The horizontal line indicates the median and the error bars indicate interquartile ranges. The horizontal dotted line in (**B**–**D**) indicates the manufacturer’s suggested cut-off. * *p* < 0.05 by Mann–Whitney test, otherwise non-significant.

**Figure 4 vaccines-12-00301-f004:**
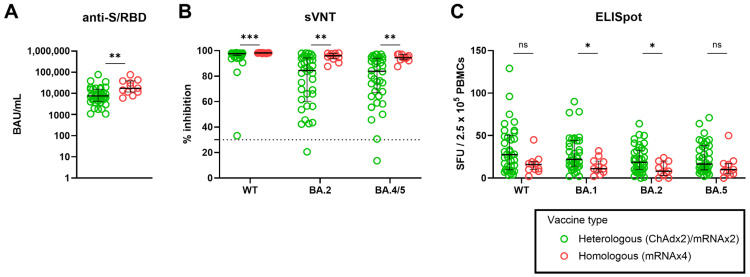
Effect of heterologous versus homologous vaccination on SARS-CoV-2–specific humoral and cellular immune response among participants with 4-exposure. Anti-S/RBD (**A**); neutralizing activity against WT, BA.2, and BA.4/5 (**B**); magnitude of T-cell responses using an enzyme-linked immunospot assay (**C**). The horizontal line indicates median and the error bars indicate interquartile ranges. The horizontal dotted line in B indicates the manufacturer’s suggested cut-off, while ns, not significant, * *p* < 0.05, ** *p* < 0.01, *** *p* < 0.001 by Mann–Whitney test.

**Figure 5 vaccines-12-00301-f005:**
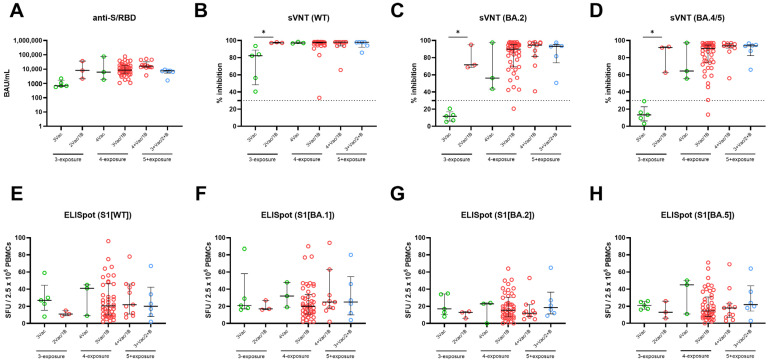
Effect of breakthrough infection and bivalent vaccination on SARS-CoV-2–specific humoral and cellular immune response, stratified by the number of total antigenic exposure. Anti-S/RBD (**A**); neutralizing activity against WT (**B**), BA.2 (**C**), and BA.4/5 (**D**); magnitude of T-cell responses using an enzyme-linked immunospot assay (**E**–**H**). The horizontal line indicates median and the error bars indicate interquartile ranges. The horizontal dotted line in (**B**–**D**) indicates the manufacturer’s suggested cut-off. Color coded as the following: green, infection-naïve; red, infected once; blue, infected two or more times. * *p* < 0.05 by Mann–Whitney test, otherwise non-significant.

**Figure 6 vaccines-12-00301-f006:**
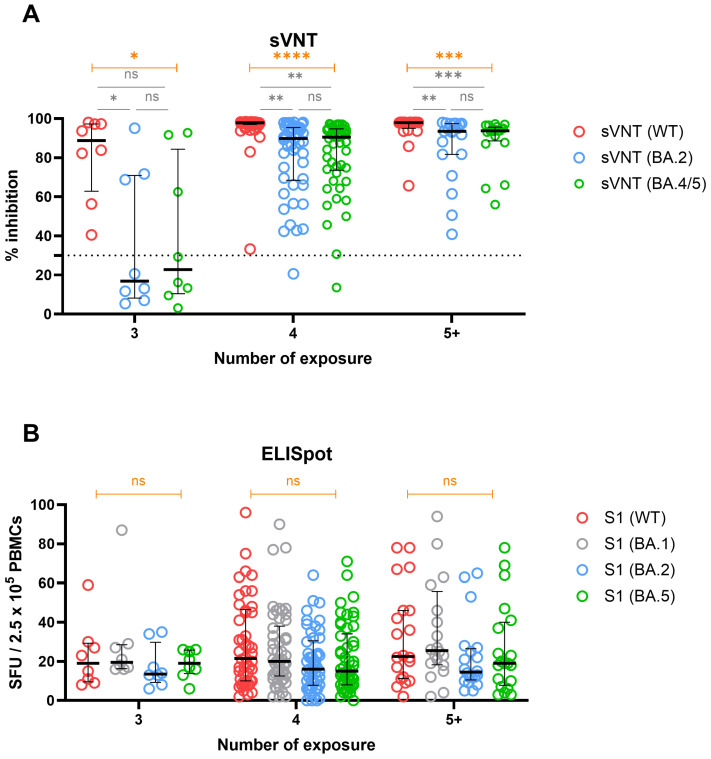
Breadth of SARS-CoV-2–specific adaptive immune response stratified by the number of total antigenic exposures. Neutralizing activity against WT and Omicron subvariants BA.2 and BA.4/5 (**A**); magnitude of T-cell responses against S1 peptide pools of various SARS-CoV-2 subvariants (**B**). The horizontal line indicates median and the error bars indicate interquartile ranges. The horizontal dotted line in A indicates the manufacturer’s suggested cut-off, while ns, not significant, * *p* < 0.05, ** *p* < 0.01, *** *p* < 0.001, **** *p* < 0.0001 by Kruskal–Wallis test (orange-colored) with post hoc Dunn’s multiple comparisons test (gray-colored).

## Data Availability

The data presented in this paper are available on request from the corresponding author.

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
