# Peer review of "Evaluation of Long-Term Adaptive Immune Responses Specific to SARS-CoV-2: Effect of Various Vaccination and Omicron Exposure"

_vaccines, 2024, doi:10.3390/vaccines12030301_

Round 1

Reviewer 1 Report

Comments and Suggestions for Authors

In this study, the authors elucidate many complex relationships regarding the evolution of immunogenicity against SARS-CoV-2 in 78 healthcare workers observed over a two-year period after receiving a two-dose vaccination regimen. The research is well-designed and should advance our knowledge of the complex relationships underlying the immune responses to SARS-CoV-2.

I endorse this work for publications after some minor corrections/suggestions right below:

. Figure 1: At the subtitle, on line 181, change “illuestrating” for “illustrating”.

. The subtitles for Figure 1 should provide clearer explanations of the image. They should describe what the vaccination status represents, the infections, why some viruses are in parentheses, etc. Personally, I find it very confusing that only the vaccination status is depicted by text, rather than a figure. It took me a while to realize that the red droplet is unrelated to the word 'booster.' I would suggest that both '2-dose' and 'booster' be represented by images as well, even if accompanied by text.

Reviewer 2 Report

Comments and Suggestions for Authors

The study investigates the impact of SARS-CoV-2 vaccine doses on infection-naive individuals longitudinally by analyzing the number of vaccines and breakthrough infections contributing to humoral immune responses and some IFN-g expressing cells at different time points in a human cohort. Results report that multiple vaccine doses, breakthrough infections, and Omicron exposure play significant roles in shaping the immune response by imprinting the immune response and tailoring the vaccine-induced immunity and it could be differ in those who received heterologous vaccines.

Please see my comment below.

I couldn't find the data included in the lines between 218 and 220. Please cite the figure or data where available in the draft.

The results shown in Fig 3A indicate...

Before discussing Figs 3B-D, let's explain the results shown in Fig 3A.

Figures 3E-H depict...? Results are missing in the draft.

It seems that no supplementary figures are attached.

Reviewer 3 Report

Comments and Suggestions for Authors

The paper presented an immunogenicity observation over a two-year period in healthy individuals who completed two-dose series and then experienced booster and/or Omicron infection. 359 infection-naive healthcare workers at Seoul St. Mary’s Hospital were included initially. The observation covered a period from March 2021 to June 2023.  Humoral immunity levels and cellular immune responses were analyzed.

However there are some major issues:

1. Supplementary Table and Figure as mentioned in the text are missing. Please provide Supplementary Table 1 and Figure 1 as mentioned in the text.

2. Please provide result descriptions for Figure 3 E-H.

3. Please check the label for Figure 5 E-F. They share the same labels.  

Minor issues are as following:

Line 26: BA 4/5 BA. 4/5

Please keep a space before and after symbols: “ = ”, “ < ” , “ > ”; for example: P<0.05 P < 0.05. Please check throughout the paper.

Line 143: 2.5x105cells/mL2.5 × 105 cells/mL

Line 148: 2.5x105 2.5 × 105

Wild type or WT, please keep consistent in the text.

Comments on the Quality of English Language

Please refer to above comments

Round 2

Reviewer 3 Report

Comments and Suggestions for Authors

No more comments.